# A Global Gene Body Methylation Measure Correlates Independently with Overall Survival in Solid Cancer Types

**DOI:** 10.3390/cancers12082257

**Published:** 2020-08-12

**Authors:** Dietmar Pils, Elisabeth Steindl, Anna Bachmayr-Heyda, Sabine Dekan, Stefanie Aust

**Affiliations:** 1Division of General Surgery, Department of Surgery, Comprehensive Cancer Center (CCC) Vienna, Medical University of Vienna, 1090 Vienna, Austria; Elisabeth.Steindl@meduniwien.ac.at; 2Department of Obstetrics and Gynecology, Comprehensive Cancer Center (CCC) Vienna, Medical University of Vienna, 1090 Vienna, Austria; Anna.Bachmayr@hotmail.com (A.B.-H.); Stefanie.Aust@meduniwien.ac.at (S.A.); 3Department of Pathology, Medical University of Vienna, 1090 Vienna, Austria; Sabine.Dekan@meduniwien.ac.at

**Keywords:** gene body, DNA methylation, CGI, overall survival, epigenetics, CpG island

## Abstract

Epigenetics, CpG methylation of CpG islands (CGI) and gene bodies (GBs), plays an important role in gene regulation and cancer biology, the former established as a transcription regulator. Genome wide CpG methylation, summarized over GBs and CGIs, was analyzed for impact on overall survival (OS) in cancer. The averaged GB and CGI methylation status of each gene was categorized into methylated and unmethylated (defined) or undefined. Differentially methylated GBs and genes associated with their GB methylation status were compared to the corresponding CGI methylation states and biologically annotated. No relevant correlations of GB and CGI methylation or GB methylation and gene expression were observed. Summarized GB methylation showed impact on OS in ovarian, breast, colorectal, and pancreatic cancer, and glioblastoma, but not in lung cancer. In ovarian, breast, and colorectal cancer more defined GBs correlated with unfavorable OS, in pancreatic cancer with favorable OS and in glioblastoma more methylated GBs correlated with unfavorable OS. The GB methylation of genes were similar over different samples and even over cancer types; nevertheless, the clustering of different cancers was possible. Gene expression differences associated with summarized GB methylation were cancer specific. A genome-wide dysregulation of gene-body methylation showed impact on the outcome in different cancers.

## 1. Introduction

Epigenetic changes are an increasingly important research field for understanding cancer biology and cancer treatment [1], despite the fact that epigenetics has not yet reached the status of a cancer hallmark [2,3]. DNA methylation on cytosines is one of the epigenetic changes which can be measured relatively easily, even genome wide, and whose impact on gene expression is—more or less—well known. In vertebrates usually most of cytosines in CpG-dinucleotides are methylated throughout the genome [4]. Only promoter regions, with or without CpG-islands (CGIs), of expressed genes are frequently hypo-methylated. One of the epigenetic changes associated with cancer (progression) is the hyper-methylation of such CGIs in promoter regions of so-called tumor suppressor genes (TSG), leading to the silencing of the corresponding genes, with the well-known examples TP53 and BRCA1. Gene body CpG-methylation is not so well understood, but usually associated with higher expression of the corresponding gene, or vice versa, hypo-methylation of gene bodies (GBs) associated with lower expression [5]. Positive correlations between active transcription and gene body methylation have recently been confirmed on the active X chromosome [6,7]. It was also shown that intragenic DNA methylation, especially if located near exon–intron boundaries, is associated with alternative splicing events [8]. Arechederra et al. discovered, in a clinically relevant hepatocellular carcinoma (HCC) mouse model, that the expression of oncogenes with hypermethylated CGIs either in their 5′-UTR or in the gene body is raised [9]. Many of these genes discovered in the mouse model—several of them well-known oncogenes—are also affected by concurrent CGI hypermethylation and gene expression upregulation in human HCC patients, especially in 56% of HCC patients, which belong to the “HCC proliferative-progenitor” subclass.

However, overall, “Intriguingly, the genes regulated by intragenic methylation in cancer cells are related to cell type-specific functions rather than tumor suppressors”, a citation from Lee et al. [8].

It was also recognized, that global hypomethylation of cancer genomes (often measured on Alu repeats [10]) is a frequent event and associated with a more aggressive tumor [11]. Here, we analyzed global CGI methylation, as well as global gene body methylation levels, in different solid cancer entities and its effect on overall survival.

## 2. Results

### 2.1. Reduced Representation Bisulfite Sequencing

DNA methylation of fresh frozen (FF) and formalin fixed paraffin embedded (FFPE) ovarian cancer tissues was measured with the enhanced reduced representation bisulfite sequencing (RRBS) methodology [12]. CpG call rates (more than five informative reads) of median 1.86 million (0.57–2.80 m) CpG-cytosines for FFPE and 2.58 million (1.53–3.44 m) for FF material were obtained. CGI coordinates were downloaded from the UCSC genome browser (HG38) and GB coordinates of protein coding genes extracted from the GENCODE V25 annotation. Beta values were calculated as ratios of methylated cytosines to total called cytosines (0 being unmethylated and 1 fully methylated) and the weighted (log_10_(calling depth + 0.1)) average of all CpG cytosines calculated for each CGI and every GB. Only averaged beta values of CGIs or GBs with at least eleven called CpGs were used for further analyses. These averaged beta values for each CGI and every gene body and sample were categorized into ten deciles, i.e., 0–10%, 10–20%, …, 90–100% and summed up.

In Appendix A the correlations of averaged GB methylation values of matched pairs of tumor tissues analyzed from FF and FFPE material are shown, indicating a high consistency of results. In addition, correlations of all possible combinations within the group of FF and FFPE tissues and between FF and FFPE tissues of different patients were obtained (Appendix A). Interestingly, there was a very high correlation between all samples, indicating a very stable GB methylation situation in ovarian cancer.

For survival analysis only values from FFPE tissues were selected as this cohort was the larger one and tissues used for DNA isolation were enriched for tumor cells using macro- and laser capture microdissection. Using the ten frequency deciles for each patient a robust survival analysis was performed with censored overall survival data as an outcome variable. In Figure 1 this procedure is outlined.

Four GB methylation deciles showed significant correlation with overall survival in ovarian cancer (red and green dots in Figure 1D), i.e., defined/extreme states (0–10%, means not methylated and 90–100%, means completely methylated) showed a negative impact on survival, whereas undefined/intermediate states (50–60% and 60–70%) showed a positive impact on survival. Using these slots a final “methylation definition factor” (MDF) was calculated as given in Figure 1E, i.e., the number of GBs with a defined/extreme methylation status (<10% and >90%) divided by the number of GBs with undefined/intermediate methylation states (50–70%), dichotomized at the optimal cutoff and used for multiple Cox regression analysis including known clinicopathologic factors such as age, FIGO stage, and residual tumor after debulking surgery (Table 1). A higher MDF, corresponding to more defined/extreme methylated genes compared to undefined/intermediate methylated genes, indicated unfavorable OS. A similar analysis with CGI methylation frequency deciles revealed no significant impact on survival at all.

A comparison of GB methylation and corresponding CGI methylation (mostly localized in the 5′-untranslated region (5′-UTR) or at the beginning of the first exon) over all samples revealed high (r > 0.8) correlations only for 4.45% of GB-CGI combinations, mainly small genes with the CGI inside their corresponding GB. In Appendix A these results are shown, in (A) the correlations between GB and CGI methylation values over all samples subdivided into ten methylation slots (GBs left plots and CGIs right plots), i.e., 0–10%, 10–20%, ..., 90–100% mean GB or CGI methylation, respectively (colored white to dark grey), in (B) the correlation of corresponding GB and CGI methylation levels averaged over all samples of all GB-CGI combinations, and in (C) the coefficients of variation (CV) of the GB methylation values (in percent) according the GB mean values, indicating more variation in low methylated GBs compared to high methylated ones (probably due to noise derived from the method used for measurement). Interestingly, the highest GB-CGI methylation correlations were observed in low methylated GBs (which showed the highest variation) and medium to high methylated CGIs.

The next question was whether the GB methylation level correlated to the RNA expression of the corresponding genes or whether genes with high GB methylation levels are globally higher expressed compared to genes with low GB methylation levels (both suggested from literature). Appendix A shows in (A) the histograms of the correlation coefficients between GB methylation values and corresponding gene expression values of all analyzed patients subdivided into ten GB methylation slots, i.e., 0–10%, 10–20%, ..., 90–100% mean GB methylation levels (colored white to dark grey) and in (B) the boxplots of log_2_ expression values of all genes in these GB methylation slots and boxplots of the corresponding gene (i.e., mRNA transcript) lengths of genes in these slots. The latter is shown, as from RNA sequencing data the expression values derived from mapped read counts are dependent on corresponding (mappable) gene lengths, if not corrected for gene lengths during bioinformatical processing. There seems to be no positive correlation between GB methylation and gene expression, neither for individual genes nor globally over all genes with the same GB methylation level. Contrary, genes with very high GB methylation levels are globally less expressed compared to genes with lower or intermediate GB methylation levels (regardless of gene length).

The next question was whether the GB methylation level correlated to the RNA expression of the corresponding genes or whether genes with high GB methylation levels are globally higher expressed compared to genes with low GB methylation levels (both suggested from literature). Appendix A shows in (A) the histograms of the correlation coefficients between GB methylation values and corresponding gene expression values of all analyzed patients subdivided into ten GB methylation slots, i.e., 0–10%, 10–20%, ..., 90–100% mean GB methylation levels (colored white to dark grey) and in (B) the boxplots of log_2_ expression values of all genes in these GB methylation slots and boxplots of the corresponding gene (i.e., mRNA transcript) lengths of genes in these slots. The latter is shown, as from RNA sequencing data the expression values derived from mapped read counts are dependent on corresponding (mappable) gene lengths, if not corrected for gene lengths during bioinformatical processing. There seems to be no positive correlation between GB methylation and gene expression, neither for individual genes nor globally over all genes with the same GB methylation level. Contrary, genes with very high GB methylation levels are globally less expressed compared to genes with lower or intermediate GB methylation levels (regardless of gene length).

### 2.2. Comparison of the Methylation Definition Factor (MDF) of Tumor Tissues, Tumor Cell Lines, and Normal Human Tissues

To compare the ratios of the defined/extreme methylation status (averaged beta values of CpGs in GBs < 0.1, i.e., unmethylated, and averaged beta values of CpGs in GBs > 0.9, i.e., methylated) to the undefined/intermediate methylation status (averaged beta values between 0.2 and 0.8), the methylation definition factor (MDF), between tumor tissues, tumor cell lines, and normal cells and tissues, RRBS data from the ENCODE consortium were downloaded and processed as above (using corresponding HG19 annotations for CGIs and GBs). In Appendix A histograms of these MDF values over all samples in each category are shown. Normal cells and tissues showed the most defined MDF values around 2 and up to 4 (with one outlier at 8, perhaps an artefact), followed by cancer cell lines with values around 1 and up to 5.5 and tumor tissues with values all below 1, regardless if measured with DNA isolated from fresh frozen or FFPE tissues. Loss of defined GB methylation, either un- or completely methylated, seems to be a universal event in cancer tissues. Comparing high and low MDF values from tumor tissues concerning overall survival revealed more aggressive tumors with high MDF values, leading to unfavorable overall survival in ovarian cancer patients. In Appendix A the distribution of all averaged GB methylation beta values with 95%-confidence intervals over all samples are shown for ovarian cancer tissues, cancer cell lines, and normal human primary cells or tissues.

### 2.3. Validation of the Impact of the MDF on Survival in other Cancer Entities

Unfortunately, the TCGA methylation data for ovarian cancer were performed with Agilent’s Human Methylation 27 bead array, comprised of about 27.6K CpGs only, which is by far not enough to validate the results (not even one CpG for each of the ~28K CGIs and ~18K protein coding GBs). Other publicly available DNA methylation data of clinical samples with corresponding survival data (RRBS, whole genome bisulfite sequencing (WGBS), or Agilent’s Human Methylation 450K array) were not available for ovarian cancer; therefore, we thought to investigate if similar correlations of MDF values with survival will be obtained for other solid cancer entities. In Table 1 cancer entities used in this study are summarized. Interestingly, in breast (BC) and colorectal (CRC) cancer comparable results were obtained using GB frequency deciles for predicting overall survival: more defined/extreme states (either highly methylated or unmethylated) showed a negative impact on survival and more undefined/intermediate states (averaged beta values around 0.5) showed a positive impact. A calculated and optimally dichotomized MDF revealed, always, an independent predictor for OS, corrected for relevant clinicopathologic factors, similar as for OvCa. In BC the MDF correlated positively with the basal subtype (Appendix A), but including the subtype information to the multiple Cox regression model even increased the hazard ratio (HR) and the significance of the dichotomized MDF (HR 2.00; *p* = 0.003). In CRC the MDF correlated positively with the CpG Island Methylator Phenotype high (CIMP-H) status (determined from TCGA colorectal cancer samples according to Hinoue et al. [18]; Appendix A) but the dichotomized MDF remained as independent predictor for OS (HR 2.07; *p* = 0.004) even when the CIMP status was added to the other clinicopathologic factors in the multiple Cox regression model. The CIMP status showed no independent impact on survival. In lung cancer, no significant association of any averaged GB methylation frequency decile with OS was revealed, therefore, no further survival analyses were performed. In pancreatic adenocarcinoma (PAAC) the impact of the averaged GB methylation frequency deciles was the other way round, more defined/extreme states (i.e., unmethylated) showed a positive impact on favorable OS and more undefined/intermediate states showed a positive impact on unfavorable OS. An optimally dichotomized high MDF, therefore, showed alone and corrected for known clinicopathologic factors (independent) significant impact of favorable OS. In glioblastoma multiforme (GBM), all deciles below averaged beta values of 0.5 (unmethylated) showed a negative impact on OS and all deciles above averaged beta values of 0.5 (methylated) revealed positive impact on OS. Therefore a “methylation-over-unmethylation factor” (MUMF) was calculated, dichotomized at the optimal cutoff, and used for OS analyses. High MUMF correlated with favorable OS, alone and corrected for (thus independent from) clinicopathologic factors. In Figure 2 SAM plots of robust Cox regressions, Kaplan–Meier estimates, and multiple Cox regression survival curves, dichotomized using the optimal cutoff, are shown.

### 2.4. Characterization of Unmethylated, Methylated, and Undefined Methylated Gene Bodies

To assess the methodological impact of the RRBS technology and the Agilent’s methylation arrays to the results and to estimate the cancer type specificity of the whole genome GB methylation state, an Isomap (a nonlinear dimensionality reduction method) and a tSNE (t-distributed stochastic neighbor embedding, a machine learning algorithm for visualization) analysis over all tumor entities using averaged beta values was performed. Appendix A shows the first four dimensions of the Isomap in three plots (dimensions 1 and 2, 2 and 3, and 3 and 4) and in Figure 3 and Figure 4 the first two dimensions of the tSNE are shown. The largest impact on the results revealed the methodology, i.e., RRBS versus methylation array, in dimension 1 of the Isomap. The clearest discrimination between cancer entities are seen in dimensions 3 and 4. In the tSNE representation all cancer entities are nearly completely separated, with OvCa samples isolated on the upper side due to methodological and biological differences. Only a few individual cancer samples are wrongly clustered, especially some lung and CRC samples in the PAAC cluster. Figure 4 shows the tSNE plots colored according the MDF/MUMF values, the optimally dichotomized MDF/MUMF, and each one of the relevant clinicopathologic or subclassification factors. To compare cancer entity specific lists of usually extreme methylated (mean averaged beta values > 0.7), usually undefined/intermediate methylated (mean averaged beta values > 0.3 and < 0.7) or usually extreme unmethylated (mean averaged beta values < 0.3) genes, Venn-like diagrams (UpSet plots) are shown in Appendix A. Interestingly, besides the differences between RRBS data (OvCa) and all methylation array data (all other cancer entities, including GBM), the overlap over all cancer entities was relatively large, indicating a robust methylation status of GBs, at least in solid cancers.

To biologically annotate genes with either preferentially unmethylated (mean averaged beta values < 0.3), methylated (mean averaged beta values > 0.7), or undefined methylated (mean averaged beta values between 0.3 and 0.7) GBs, lists over all samples per cancer entity where built averaging all averaged beta values per single gene and trichotomized according mean beta cutoffs of 0.3 and 0.7. In Figure 5 gene ontology (GO) and pathway annotations for each category (methylated > 0.7, undefined between 0.3 and 0.7, and unmethylated < 0.3) of every cancer type with GB methylation impacts on OS are shown as word clouds in colors indicating enrichment *p*-values.

To assess the consistency of the GBs in the mean GB methylation slots, i.e., high methylated (>0.7), intermediate methylated (≥0.3 and ≤0.7), and low methylated (<0.3), Appendix A shows the distribution of the percentages of all genes consistently annotated to one of these three slots. Interestingly, high percentages of GBs in the same slot are only seen for intermediate methylated GBs in all cancer entities. Therefore, there are only few genes with consistent high or low GB methylation values if cutoffs are applied, despite the fact that GB methylation values correlated substantially over all samples (cf. Appendix A). To analyze and annotate the overlap of consistently high, intermediate, or low methylated GBs, Appendix A show the overlap of gene (i.e., GB) lists which are in >90% of samples annotated to the respective GB methylation slot, thus consistently categorized, and the annotations of these gene lists in GO enrichment plots, respectively.

### 2.5. Correlation of the MDF and MUMF with Gene Expression

Genes whose expressions were significantly correlated to the MDF in OvCa, BC, CRC, and PAAC and to the MUMF in GB were determined from RNA-sequencing data, either from our own data, OvCa [19,20], or from the TCGA database for all other cancer entities. In OvCa (false discovery rate (FDR) < 10%) 1039 genes were positively correlated with the MDF and 730 were negatively correlated; in BC (FDR < 5%) 3177 and 2920 genes, in CRC (<5%) 1405 and 1179 genes, and in PAAC (<5%) 1780 and 2052 genes, respectively. In GB (FDR < 10%) 339 genes were positively correlated with the MUMF and 505 negatively. Overlaps of significantly differentially correlated genes are shown in Appendix A, indicating nearly no relevant overlaps between tumor entities. There were completely no overlaps of significantly positively or negatively correlated genes over all samples, even if GBM was excluded. Only two genes were positively correlated in OvCa, BC, CRC, and negatively correlated in PAAC, namely SLC7A11 and TFAP2A, and only one gene was negatively correlated in OvCa, BC, and CRC and positively correlated in PAAC, namely LIMS2. The rational for this combination was that the MDF showed an inverse impact on OS in PAAC compared to OvCa, BC, and CRC.

A Signaling Pathway Impact Analysis (SPIA) with lists of significantly with the MDF/MUMF factor associated genes revealed for three cancer entities significant (FDR < 5%) deregulated KEGG pathways (total 202 KEGG pathways considered, Appendix A), for OvCa two activated ones: “Systemic lupus erythematosus” (FDR 0.08%, Appendix A) and “Staphylococcus aureus infection” (FDR 0.5%), for BC 30 inhibited pathways: “Cytokine-cytokine receptor interaction” (Appendix A), “Neuroactive ligand-receptor interaction”, “Chemokine signaling pathway”, “Natural killer cell mediated cytotoxicity”, “PI3K-Akt signaling pathway”, “Systemic lupus erythematosus”, “Pathways in cancer”, “Staphylococcus aureus infection”, “Th1 and Th2 cell differentiation”, “Th17 cell differentiation”, “Focal adhesion”, “Measles”, “NF-kappa B signaling pathway”, “Human T-cell leukemia virus 1 infection”, “Necroptosis”, “Inflammatory bowel disease (IBD)”, “Complement and coagulation cascades”, “Leukocyte transendothelial migration”, “Hepatitis B”, “Adipocytokine signaling pathway”, “Autoimmune thyroid disease”, “Allograft rejection”, “Malaria”, “Asthma”, “Intestinal immune network for IgA production”, “Rap1 signaling pathway”, “Osteoclast differentiation”, “Antigen processing and presentation”, “ECM-receptor interaction”, FDRs from 10^−8^% to 4.9%, and for GB one inhibited “Taste transduction” and one activated “Choline metabolism in cancer” pathway (both FDR 2.5%, Appendix A) (cf. Appendix A). Interestingly, no pathway regulation direction combination overlapped between these cancer entities. For CRC and PAAC no significant KEGG pathways were revealed at all (SPIA results tables are given in Appendix A).

For further biological annotations lists of significantly (false discovery rates <5% or <10% for OvCa and GB) positively and negatively with the MDF/MUMF correlated genes for each cancer entity were composed and annotated as above. In Figure 6 these are shown as word clouds in colors indicating enrichment *p*-values.

## 3. Discussion

DNA methylation is an important event and factor in epigenetic gene expression regulation [21]. Globally, CpG dinucleotides are substantially under-represented in the human genome, except in a few thousand short regions (few hundred to thousand bases long) with enriched CpG-frequencies, known as CpG-islands (CGIs). Non-CGI CpGs are usually methylated in the human genome and CGI CpGs unmethylated in association with active genes. Inactive genes are characterized by methylated CGIs in promoter regions, if a CGI is present at all. Therefore, CGIs are usually involved in gene regulation by a regulator protein (e.g., transcription factors) binding, either at promoter regions or in enhancers, usually following the logic, higher methylation lower regulator protein binding and expression and vice versa. Globally, solid cancers are characterized by global genome-wide hypo-methylation, i.e., usually methylated CpGs get less methylated in cancer tissues, and hyper-methylation of specific CGIs, especially in front of so-called tumor suppressor genes (TSGs). These TSGs are specifically silenced by hyper-methylation of corresponding CGIs to allow the tumor a more aggressive, malignant, or metastasizing phenotype.

GB methylation, i.e., the methylation status of CpGs throughout the GB, including introns and exons, is a more unclear factor. Usually it is considered that a higher GB methylation is correlated with higher gene expression [5,7], but the concrete mechanisms are not clear. Only the inhibition of spurious transcription initiation of GB methylation of high expressed genes was shown in mouse stem cells [22]. GB methylations at specific regions are also correlated to splicing [8,23].

Using a global approach of correlating GB methylation levels with expression of corresponding genes revealed no proof of a positive correlation. Only genes with high GB methylation levels showed a slightly positive median correlation coefficient, but genes with low and intermediate GB methylation levels showed even a slightly negative median correlation coefficient (Appendix A).

This work is to our knowledge the first report of the analysis of a global GB methylation status with impact on overall survival in several solid cancer entities. Using averaged methylation beta values of GBs allows near perfect classification of the cancer entities analyzed in this work (Figure 3 and Appendix A), even as GB methylation values were highly correlated over samples and types of tissues, fresh frozen or FFPE (Appendix A), and lists of differentially methylated GBs were rather similar over cancer entities (Appendix A), indicating a similar GB methylation status of individual genes over tissues. There were only few genes where GB methylation correlated to the corresponding CGI methylation (Appendix A) or to gene expression (Appendix A). Only genes with highly methylated GBs were less expressed as a group compared to genes with lowly or intermediately methylated GBs, even considering gene length as possible bias for expression analysis from RNA sequencing data. Nevertheless, a positive impact of a globally more defined/extreme state of GB methylation (i.e., more GBs either highly or lowly methylated compared to GBs with an in-between methylation status), represented by a methylation definition factor (MDF) on unfavorable OS, was initially shown with our own RRBS data in OvCa (Figure 1) and validated with TCGA data in breast (BC) and colorectal (CRC) cancer (Table 1 and Figure 2). Even the exclusive analysis of intronic and exonic regions of the GBs revealed similar results in OvCa compared to the analysis of the complete GB, i.e., a significant impact on OS (cf. Table 1). Interestingly, in pancreatic adenocarcinoma (PAAC), the impact on OS was the other way round, i.e., more defined/extreme methylated GBs correlated with favorable OS (Figure 2). All OS impact results were significant and independent, i.e., even if corrected for cancer type specific clinicopathologic factors and for subtype in BC and for the CpG Island Methylator Phenotype (CIMP) in CRC (even as the MDF was positively correlated to the basal subtype in BC and the CIMP-high status in CRC). For lung cancer no correlation of GB methylation with OS was found (even with a large patient cohort) and for glioblastoma the number of methylated GBs (averaged beta values > 0.5) compared to the number of unmethylated GBs (averaged beta values < 0.5) correlated with favorable OS–methylation-over-unmethylation factor (MUMF)—alone and corrected for clinicopathologic factors. Glioblastoma is not a carcinoma (i.e., derived from epithelial cells) as the other analyzed cancer entities above, but derived from astrocytes, with a rather different (cancer) biology. Therefore, the different impact of GB methylation on OS is not surprising.

Our interpretation of the absence of significant correlations of single GB methylation levels with OS but a highly significant summarized GB methylation measure (i.e., the ratio of high/low methylated to intermediate methylated GBs, described here as MDF and MUMF) is that this measure indicates a malignant epigenetic phenotype dysregulating the whole transcriptional process in the cell but not tumor suppressor genes or oncogenes predominantly. However, this global dysregulation of expression is difficult to analyze with typical RNA-sequencing experiments, as the absolute expression levels of genes in a cell are completely unknown and even more unknowable, only relative differences of expressions between genes and samples can be determined. Probably the epigenetic state described here also influences global splicing events [8], even splicing events leading to circular RNAs, known to play a role in cancer [24]. To analyze epigenetically driven splicing, not poly(A)-enriched and much deeper sequenced cohorts of tumors would be necessary.

The impact of the MDF/MUMF on gene expression is completely different for all cancer entities with insignificant overlap on gene level and no overlap on dysregulated pathway level at all. Only SLC7A11, TFAP2A, and LIMS2 are commonly regulated by the MDF in carcinomas (not in GBM), but only if the direction of the correlation was inversed for PAAC, whose impact on OS was also inverse. A pathway enrichment analysis also revealed very different results from all cancer entities, with either many significantly dysregulated pathways for BC, only a few significantly dysregulated pathways for OvCa and GMB, or even no significantly dysregulated pathways for CRC and PAAC. Therefore, we want cite again the publication from Lee et al. [8] directly, “Intriguingly, the genes regulated by intragenic methylation in cancer cells are related to cell type-specific functions rather than tumor suppressors” and want add “... or oncogenes”, which seems to be proven again with our results.

The upside down impact of GB methylation in PAAC compared to OvCa, BC, and CRC remains interesting, as does the complete independence of global GB methylation—as analyzed here—and OS in lung cancer, even with a rather large cohort of patients.

## 4. Materials and Methods

### 4.1. Aim

Aim of this study was to examine the impact of global methylation states, i.e., global CpG islands (CGI) methylation and global gene body (GB) methylation measures, on the outcome of patients with solid cancers.

### 4.2. Patient Cohorts

Reduced representation bisulfite sequencing (RRBS) was performed with DNA isolated from 25 fresh frozen and 45 formalin-fixed and paraffin-embedded (FFPE) high grade serous ovarian cancer tumor tissues. The 25 samples were a subset of the 45 cohort. In total all tissue samples were from 45 patients with diagnosed high grade (grade 2 or 3), late stage (FIGO III or IV), and serous epithelial ovarian cancer which were characterized in more detail in several publications [19,20,24,25,26,27,28,29,30]. Nineteen patients were already deceased. All patients signed an informed consent for providing tissues to this study according the ethical review board (Ethics Committee of the Medical University of Vienna) Nos. 366/2003, 793/2011, and 1076/2018.

### 4.3. DNA Isolation and Reduced Representation Bisulfite Sequencing (RRBS)

Total DNA from fresh frozen tumor tissues (frozen and stored in the gas phase of liquid N_2_) was isolated with the DNeasy Blood and Tissue Kit (QIAGEN, Venlo, Netherlands) and from FFPE slices after macro- (~80%) or laser capture microdissection (~20%, depending on the size and uniformity of the epithelial tumor areas) of tumor tissues with the AllPrep DNA/RNA FFPE Kit (QIAGEN). DNA amount was measured by NanoDrop™ 8000 Spectrophotometer (Thermo Fisher Scientific, Waltham, MA, USA) and the PicoGreen^®^ dsDNA Quantitation Reagent (Promega, Madison, WI, USA) and the quality was assessed by the Agilent High Sensitivity DNA Kit on the 2100 Bioanalyzer (Agilent Technologies, Santa Clara, CA USA). RRBS was performed and methylation status of CpGs called from 100 ng total DNA according to the protocols published by the Biomedical Sequencing Facility (BSF) in Vienna [12,31].

### 4.4. Gene Body and CpG Island Analysis

Gene body (GB) of protein coding genes and CpG island (CGI) coordinates for HG38 and HG19 human genomes were obtained from the UCSC genome browser and the GENCODE V25 annotation. From our RRBS data from high grade serious ovarian cancer, the average beta values (i.e., the methylation level using the ratio of methylated bases to all called bases at a specific CpG position) of all CpGs mapping to either a CGI or the GB of a protein coding gene was calculated (resulting in one average beta value per each CGI and one average beta value per each GB for each sample). The corresponding sequencing depth (log_10_(calling depth + 0.1)) for each CpG was weighted in the calculation of the beta values. Similarly, average beta values for GB were calculated using publicly available TCGA beta values without weighting if at least eleven beta values of CpGs were available for the GB. Beta values and clinicopathologic data from TCGA cohorts were obtained with R-package TCGA2STAT 1.2. For every sample the sum of all GBs and CGIs whose average beta value maps to the deciles (0–10%, 10–20%, ..., 90–100%) were summed up and the sums of all deciles for each sample used for correlation to overall survival. The procedure is summarized in Figure 1. With this approach a measurement for the global methylation status in CGI and GBs was introduced which was further used in the subsequent survival analyses.

### 4.5. Overall Survival Analysis of Gene Body and CpG Island Decile Numbers

To assess the impact of the global methylation status (measured as sum of GBs or CpG islands in each decile) on overall survival (OS, censored data) the robust (permutation based) Significance Analysis of Microarrays (SAM) method (R-package samr 3.0 [32]) was used with a false discovery rate (FDR) cutoff of 5%. Only the deciles which were significantly associated with OS were considered for the next step, the calculation of a methylation definition factor (MDF) for each sample for carcinomas (OvCa, BC, CRC, and PAAC): by dividing the number of unmethylated (e.g., OvCa: 0–10% decile) or methylated (e.g., OvCa: 90–100% decile), thus extreme or defined methylated, GBs by the number of undefined or intermediate methylated (e.g., OvCa: 50–60% and 60–70% deciles) GBs. The concrete used deciles and formulas for each cancer entity are given in Table 1. For glioblastoma a simple methylation-over-unmethylation factor (MUMF) was calculated, dividing all numbers of methylated GBs (averaged beta values > 50%) by all numbers of unmethylated GBs (averaged beta values < 50%), because all deciles over 50% showed impact on OS in one direction and all deciles below 50% in the opposite direction (*cf*. Table 1). To determine the optimal (most informative) cutoff to dichotomize the MDF or MUMF, a Cox regression model was built including all relevant and available clinicopathologic factors, thus considering in the model (shown in Table 1), and the cutoff with the lowest *p*-value determined using the ‘cut-p’ function from R-package survMisc 0.5.5. Finally, patients were dichotomized according the respective optimal cutoff and single and multiple Cox regression analyses performed, the latter with subsequent stepwise backward selection of factors optimizing the Akaike information criterion (AIC) using function stepAIC from R-package MASS 7.3-51_4 [33]. Factors included together with the optimally dichotomized MDF or MUMF in the final Cox models are indicated by names without parentheses in Table 1. Kaplan–Meier estimates and representations of the final multiple Cox models are shown in Figure 2.

### 4.6. Annotation of Genes with Different Methylation Status and Genes Which Expressions Correlated with the MDF or MUMF

Isomap and T-distributed Stochastic Neighbor Embedding (tSNE) analyses were performed by R-packages RDRToolbox 1.34.0 (function ‘Isomap’) and Rtsne 0.15 (function ‘Rtsne’ with parameters: ‘pca = FALSE, perplexity = 30, theta = 0.5, dims = 2’) [34], respectively. For biological annotation of genes with different averaged methylation beta values for each cancer entity three lists were composed, one with genes of mean (over all samples of one tumor type) < 0.3 averaged GB methylation beta values (Unmethylated), one with genes of mean > 0.7 averaged GB methylation beta values (Methylated) and the third list with all remaining genes (Undefined). Overlap of these lists were determined and presented in Venn-like UpSet diagrams, prepared by the R-package UpSetR 1.4.0 (function *upset*) [35]. Furthermore, these gene lists were analyzed for enrichments in gene ontology (GO) terms (biological processes, cellular components, and molecular functions) and KEGG and Reactome pathways with function gosummaries from R package GOsummaries 2.22.0 [36].

For the correlation of the MDF or MUMF with gene expression the corresponding RNA-sequencing data were used, for OvCa own data [20] and for all other cancer entities the TCGA data, downloaded again with the TCGA2STAT R-package. Expressed genes were filtered using 0.5 counts per million in halve of samples and remaining raw read counts were normalized and weighted according to quality with R-function ‘voomWithQualityWeights’ from R-package limma 3.40.6 [37] (using the cyclicloess normalization, i.e., cyclicly applying loess normalization). These genes were correlated to the corresponding MDF or MUMF values and gene lists composed of significantly positively or negatively correlated genes (for OvCa and GBM with an FDR cutoff of 10% and all other cancer entities of 5%). Overlap of these gene lists are shown in Appendix A using the upset method as above. GO and pathway enrichments of these gene lists were analyzed as above. Differentially regulated KEGG-pathways were determined by a combined gene over-representation and perturbation analysis using Signaling Pathway Impact Analysis (SPIA), implemented in the Bioconductor R-package SPIA v2.38.0 [38], with 10,000 permutations and the normal inversion “norminv” *p* combination method. Correction for multiple testing was done by the False Discovery Rate (FDR) method (Benjamini-Hochberg). Significant pathways were illustrated with the Bioconductor R-package pathview v1.26.0 [39].

## 5. Conclusions

Global gene body methylation measures, in carcinomas the quotient of numbers of defined/extreme methylated GBs (either nearly complete methylated or unmethylated) to numbers of undefined/intermediate methylated GBs—the methylation definition factor (MDF)—correlated significantly with overall survival in ovarian cancer, breast cancer, colorectal cancer, and pancreatic cancer, but not in lung cancer. This impact was always independent from known cancer specific clinicopathologic factors and in the case of breast and colorectal cancer also from the molecular subtype or the CpG island methylator phenotype (CIMP) status, respectively (nevertheless, the MDF was correlated to both factors). Interestingly, in ovarian cancer, breast cancer, and colorectal cancer the impact of more defined/extreme methylated genes was an unfavorable predictive factor and in pancreatic cancer the other way round. A similar “definition” factor of CpG islands showed no such impact on OS, at least in ovarian cancer.

In glioblastoma the quotient of numbers of methylated GBs to numbers of unmethylated GBs—the methylation-over-unmethylation factor (MUMF)—was a significant and independent predictor, if high for unfavorable OS.

## Figures and Tables

**Figure 1 cancers-12-02257-f001:**
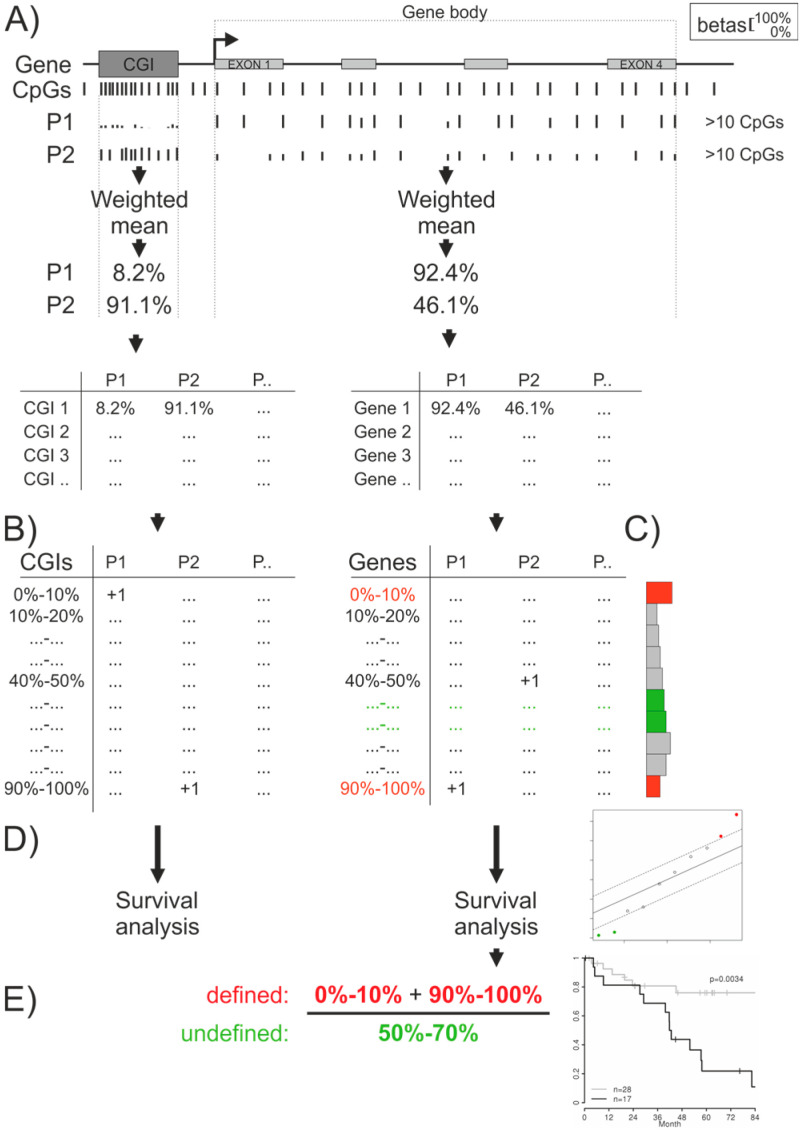
Overview of the generation of the global CpG island (CGI) and gene body (GB) methylation measures used for correlation with overall survival (OS). (**A**) Beta values from CpGs mapped to CGIs or GBs were averaged—from RRBS data with a weighted method—and (**B**) summarized in decile slots, (**C**) shown as a histogram exemplarily for the GBs of one patient. (**D**) Using these summarized decile numbers robust Cox regression analyses with censored OS data were performed (plot on the right showing the expected and observed scores for the ten deciles and the 5% false discovery rate (FDR) cutoff as dashed line; in red and green the significant deciles). (**E**) Using all negatively (red dots in the plot of (D)) and positively (green dots in the plot of (D)) with OS correlated deciles a methylation definition factor (MDF) was calculated as indicated and used for Cox regression analysis including known cancer type specific clinicopathologic parameters. An optimal cutoff for the MDF was determined from this model and a Kaplan–Meier estimate using the optimally dichotomized MDF is show on the right. A plot of the Cox model, corrected for clinicopathologic factors, is shown in Figure 2 and details of the Cox regression models are given in Table 1.

**Figure 2 cancers-12-02257-f002:**
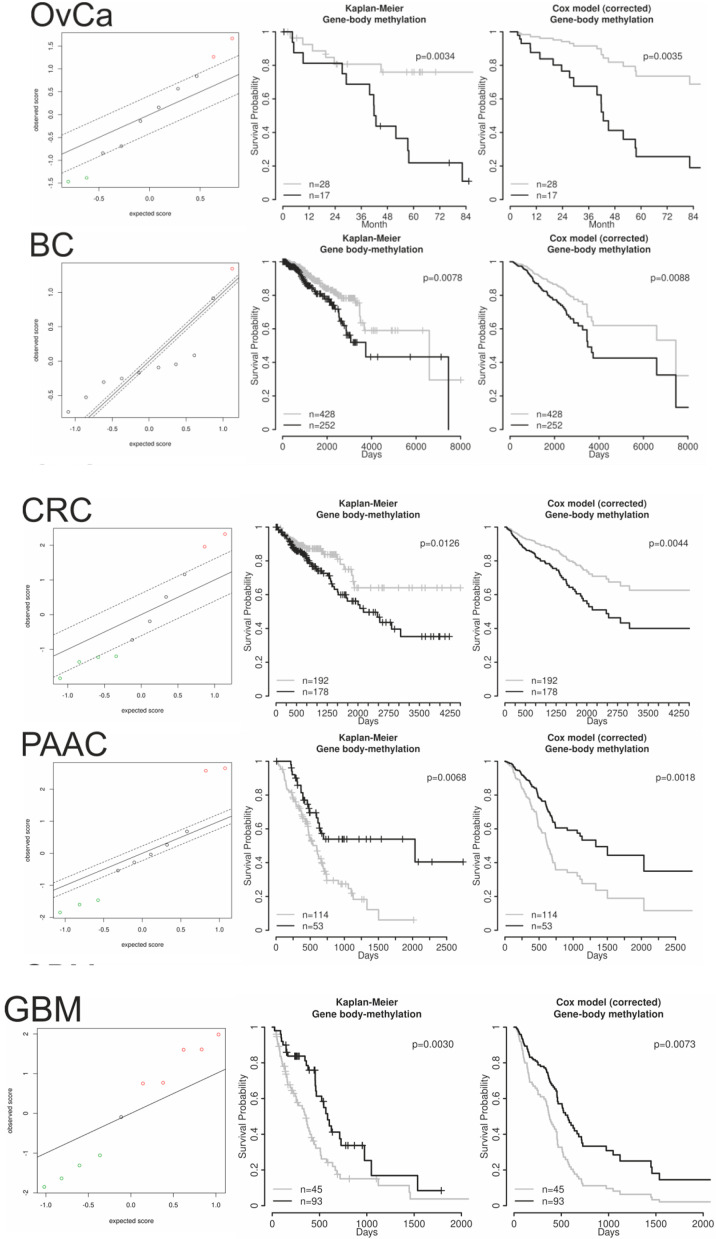
Significance Analysis of Microarray (SAM) plots of Cox regressions (X-axis, expected score; Y-axis, observed score; dashed line, threshold cutoff at 5% false discovery rate, FDR; red dots, associated with unfavorable overall survival (OS); green dots, associated with favorable OS), Kaplan–Meier estimates of the optimally dichotomized MDF/MUMF, and survival curves of the corrected Cox regression models (details are given in Table 1). (No censored patients are indicated for the latter plots, as these are survival estimates from multiple Cox regression models).

**Figure 3 cancers-12-02257-f003:**
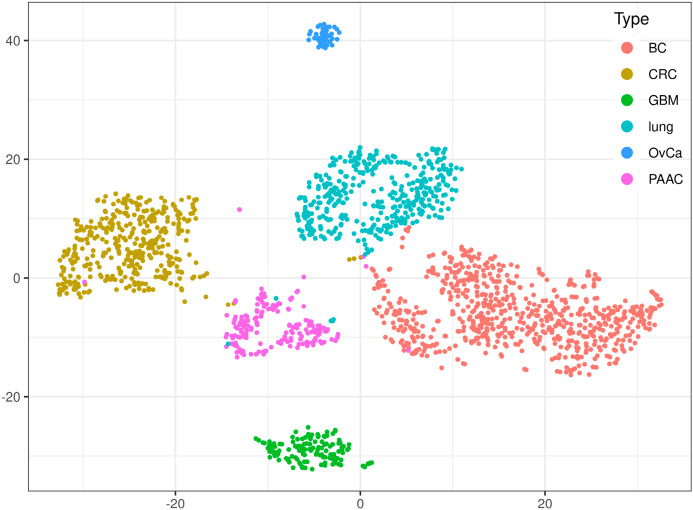
T-distributed stochastic neighbor embedding (tSNE) of the GB-averaged beta values of all cancer entities (*n* = 7.403, measured from all cancer types).

**Figure 4 cancers-12-02257-f004:**
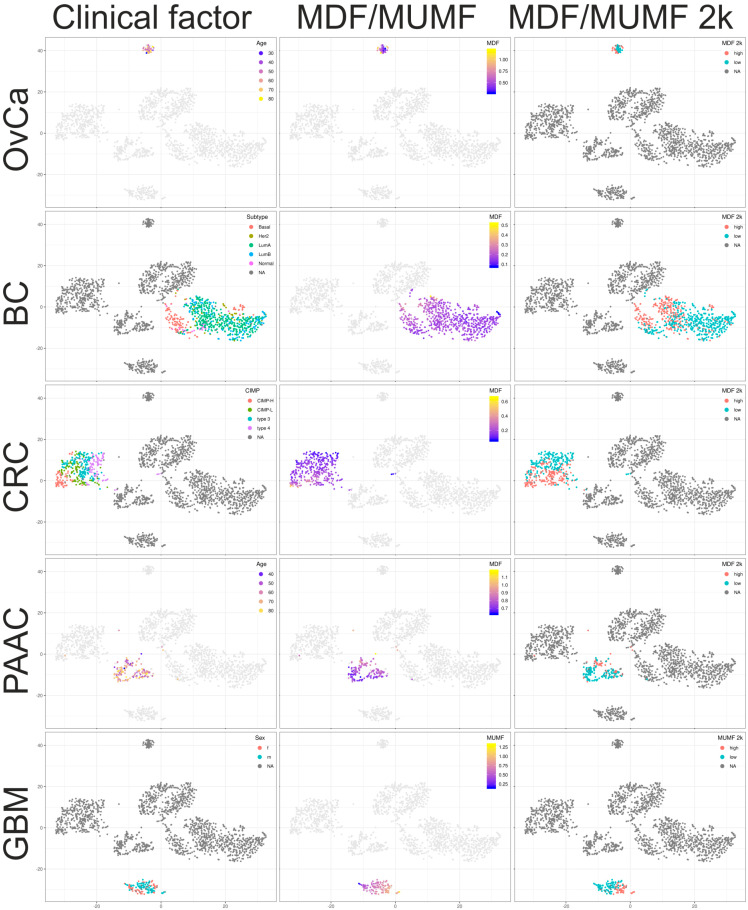
T-distributed stochastic neighbor embedding (tSNE) of the GB-averaged beta values as in Figure 3, colored according one relevant clinicopathologic factor (age for OvCa and PAAC, the subtype for BC, the CIMP status for CRC, and sex for GBM) in the left panel, colored by the MDF/MUMF values in the middle panel, and colored according the optimally dichotomized MDF/MUMF in the right panel.

**Figure 5 cancers-12-02257-f005:**
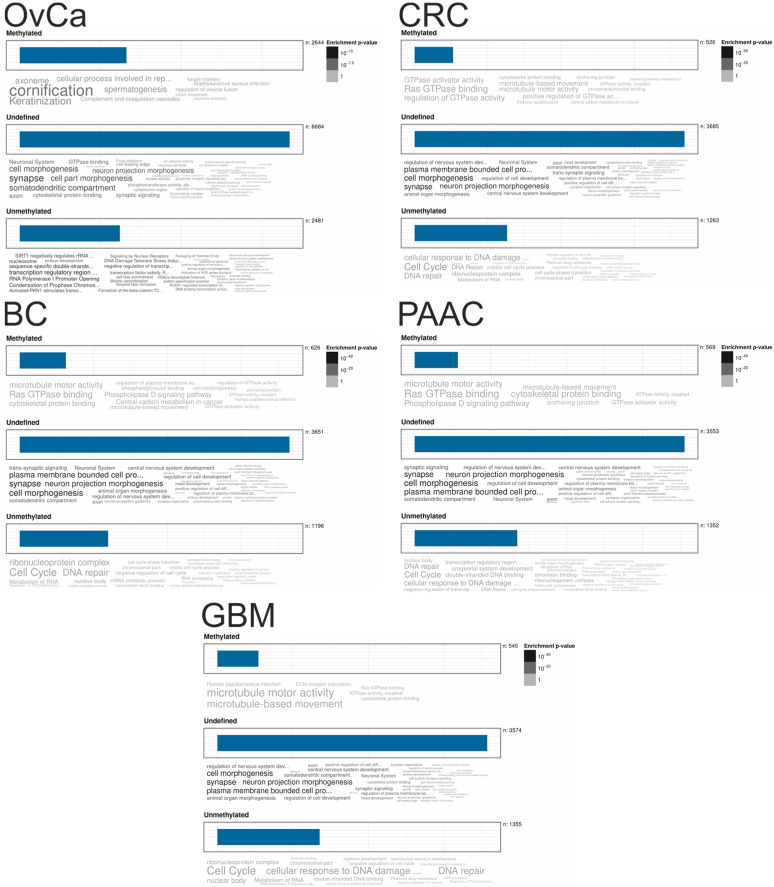
Enrichment plots of gene ontology terms and pathways are shown for lists of genes preferentially (averaged over all samples from each cancer type) methylated (averaged betas > 0.7), unmethylated (averaged betas < 0.3) or undefined methylated (averaged betas between 0.3 and 0.7). Enrichments are indicated in word clouds by size and color (*p*-values). Overlap of gene lists are shown in Appendix A.

**Figure 6 cancers-12-02257-f006:**
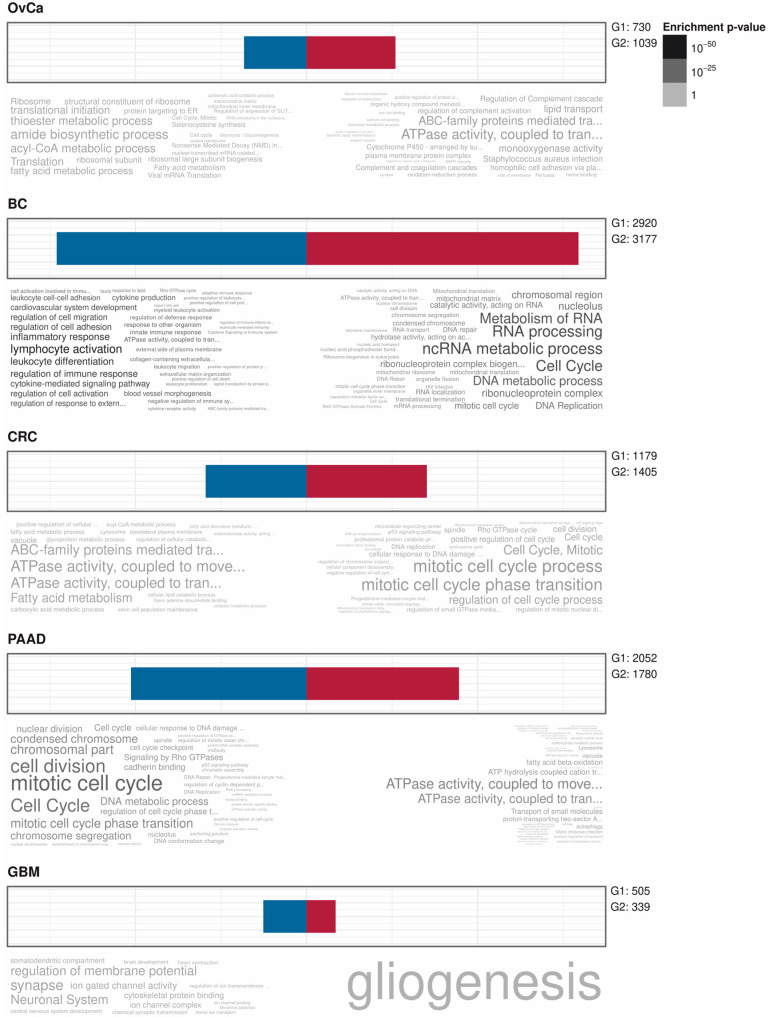
Enrichment plots of gene ontology terms and pathways of negatively (blue; G1) and positively (red; G2) with the MDF/MUMF correlated genes in different cancer entities are shown. For OvCa and GBM a 10% FDR cutoff, for all other cancer entities a 5% cutoff was used. Enrichments are indicated in word clouds by size and color (*p*-values). Overlaps of gene lists are shown in Appendix A.

**Table 1 cancers-12-02257-t001:** Overview of Characteristic Numbers (Samples, Death Event Numbers, Significant Deciles, Dichotomized Numbers, Hazard Ratios, and *p*-Values of Single and Multiple Cox Regression Analyses and Included Clinicopathologic Factors) of Cancer Types Analyzed in This Study (the grey shadow indicates subanalyses for ovarian cancer and is otherwise alternating for better readability).

Characteristics and OS Analyses						Cox Regression (Overall Survival)			
Cancer Type	Patients ^1^	Gene Bodies ^2^	SAM Sign.^3^	MDF Predictor (Def/UnDef) ^4^	Cutpoint ^5^	Univariate	Multiple	Clinical Params ^6^	Cit
Ovarian (OvCa)	45 (19)	17,798	+ 0–10% + 90–100% −50–60% −60–70%	D: <10% and >90% U: 50–70%	28 vs. 17	HR 4.29 (1.62–11.3) *p* = 0.0034	HR 4.44 (1.63–12.08) *p* = 0.0035	Age Residual tumor (FIGO stage) (Grade)	RRBS
Ovarian—Exons	45 (19)	14,902 (Exons)	+ 0–10% + 90–100% −all other deciles	D: <10% and >90% U: 30–70%	27 vs. 18	HR 3.17 (1.24–8.11) *p* = 0.0162	HR 3.44 (1.31–9.06) *p* = 0.0124	Age Residual tumor (FIGO stage) (Grade)	RRBS
Ovarian—Introns	45 (19)	15,223 (Introns)	+ 0–10% + 80–90% +90–100% −60–70% −70–80%	D: <10% and >80% U: 60–80%	28 vs. 17	HR 5.33 (1.96–14.52) *p* = 0.0011	HR 6.10 (2.01–18.51) *p* = 0.0014	Age Residual tumor (FIGO stage) (Grade)	RRBS
Breast (BC)	690 (87) 680 (84)	7814	+ 10–20%	D: <20% U: 30–70%	428 vs. 252	HR 1.79 (1.17–2.76) *p* = 0.0078	HR 1.78 (1.16–2.75) *p* = 0.0088	Stage (Histology)	[13]
Colorectal (CRC)	390 (87) 370 (80)	7816	+ 70–80% + 80–90% −20–30% −30–40% −40–50% −50–60%	D: >70% U: 20–60%	192 vs. 178	HR 1.8 (1.13–2.85) *p* = 0.0126	HR 1.96 (1.23–3.11) *p* = 0.0044	Stage (Sex) (Histology) (Site)	[14]
Pancreatic (PAAC)	184 (99) 167 (91)	7816	−0–10% −10–20% −20–30% + 40–50% + 50–60%	D: <30% U: 40–60%	114 vs. 53	HR 0.44 (0.27–0.74) *p* = 0.0018	HR 0.49 (0.29–0.82) *p* = 0.0068	Age Residual tumor Radio therapy (Sex) (Stage)	[15]
Lung	238 (150)	7815	n.s.	n.d.^7^	n.d.	n.d.	n.d.	n.d.	[16]
Glioblastoma (GBM)	138 (93) 126 (82)	7816	+ 0–10% + 10–20% + 20–30% + 30–40% + 40–50% −60–70% −70–80% −80–90% −90–100%	Methylated/Unmethylated (MUMF) M: >50% UM: <50%	76 vs. 50	HR 0.49 (0.31–0.79) *p* = 0.0030	HR 0.50 (0.30–0.83) *p* = 0.0073	Sex Histology Radio therapy	[17]

^1^ Number of patients with survival data (death events) (Number of patients with complete clinical data (Death events) used for multiple Cox regression, if not all). ^2^ Number of gene bodies with >10 CpG beta values. ^3^ Significant frequency deciles with FDR < 5%. “+” means positive impact on unfavorable OS and “−” vice versa. n.s., not significant. ^4^ Definition of the methylation definition factor (MDF) and the methylation-over-unmethylation factor (MUMF) as predictor for overall survival. ^5^ Dichotomization using the optimal cut-point of the MDF/MUMF predictor determined with multiple Cox regression models including all clinical parameters (high MDF/MUMF vs. low MDF/MUMF). ^6^ Clinical parameters finally included into the multiple Cox regression model together with the dichotomized MDF predictor, determined by minimizing the Akaike Information Criterion, AIC (in parentheses, additional clinical parameters considered for the Cox model which were excluded). ^7^ n.d., not determined.

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
