# Peer review of "A Global Gene Body Methylation Measure Correlates Independently with Overall Survival in Solid Cancer Types"

_cancers, 2020, doi:10.3390/cancers12082257_

Round 1

Reviewer 1 Report

In this manuscript, Pils and collaborators obtain the DNA methylation profiles of 45 ovary cancer samples and analyze the correlation of DNA methylation at promoters and gene bodies with overall survival of the patient. The authors observe a significant negative correlation between extreme methylation patterns at gene bodies (highly and lowly methylated) and survival, whereas intermediate methylation profiles show a positive impact on survival. The authors, next, obtain publicly available data on the methylation profile of normal and different tumor tissues and cancer cell lines to perform the same analyses, and observe a variable relationship between DNA methylation at gene bodies and survival. Finally, the authors perform additional expression and gene ontology analyses of the genes associated to the different outputs and methylation states.

Overall, the manuscript is well presented and the conclusions are supported by the experiments and analyses performed by the authors. The work is mainly descriptive, but the analysis of the "defined" vs. "undefined" methylated GBs is interesting, and the relationship between different degrees of DNA methylation with survival is innovative.

Still, I think that, prior to publication, the authors should consider to perform some additional analyses and revise some sections of the manuscript.

First, I think that there are some parts of the text that need to be extensively revised. The abstract, for instance, is unclear. It should provide a brief overview of the full manuscript, avoiding details such as technical specifications and p-values. The introduction is not very extensive, and I miss some references to papers describing the relationship between DNA methylation at gene bodies and cancer, such as https://doi.org/10.1038/s41467-018-05550-5. Also, along the results section, some technical details could be moved to the methods section, such as the normalization of the transcriptomics data, in section 2.5.

In the first section of the results, it is unclear to me how the samples were obtained. According to the methods, 25 flesh frozen and 45 FFPE fixed samples were processed. However, the 25 flesh frozen samples were a subset of the 45 fixed ones. Why these samples were processed twice? Are there differences between frozen and fixed samples?

At the end of the first section, the authors state that methylation of CGIs do not show any correlation with prognosis, but how is the relationship between the methylation of CGIs and GBs in the genes that show correlation between GBs methylation and survival?

The correlation between extreme/intermediate methylation profiles and disease prognosis is indeed interesting. However, I miss some information. For instance, for the particular case of ovarian cancer, that is the data generated by the authors, from the 45 patients, how is the overlap of the methylation pattern of the GBs? I mean, are there GBs that are consistently highly methylated, GBs that are consistently lowly methylated and GBs that always show an intermediate pattern? And if these is the case, what are these genes and what is their relationship to the development of ovarian cancer? And the same for the other cancer types.

The results are not very consistent between the different cancer types. What is the explanation of the authors for this observation? Also, in the comparison between the solid tumors, to me it is unclear why the correlation between extreme/intermediate methylation states and prognosis is so different between cancer types but, when comparing the GBs showing different methylation status in Supplementary Figure 7, they show a strong overlap, even with lung sample, that does not show any correlation, and with pancreas, for which the correlation is the opposite. Are these GBs in Figure S7 associated to good/bad prognosis, or are they all GBs within each sample? How is the methylation status of the normal tissues over these genes? I think that the authors should compare the GBs that show consistently high/intermediate/low levels of methylation between the different cancer types, as well as the GBs that are specific for each cancer to try to uncover the different origin of each tumor.

Also, when comparing the high/low methylated gene bodies with the expression of these genes, I miss some analyses. For example, how is the correlation between methylation and expression in each case? I mean, is it consistent with previous reports that show higher levels of methylation correlate with higher levels of expression? The results from the expression section are not very conclusive, and I think that they need to be further discussed in the manuscript.

Reviewer 2 Report

The article entitled "A global gene body methylation measure correlates independently with overall survival in solid cancer types" is considered for publication.

This article addresses a current topic with impact in cancer. Thus, is my opinion that the article is suitable for publication. The authors comply with their aim, so I only have minor comments. The article is easy to read and the information is clear and straight to the point.

In my opinion the introduction section is professionally written, however it has a lack of references. I understand that there are few references regarding this issue, but the authors should perform an exhaustive article search in order to justify the work here presented. As an example, Arechederra et al. 2018 published an interesting article showing that hypermethylation of gene bodies is predictive of elevated oncogene levels in cancer. Might be a good article to refer in the introduction and discussion. (Hypermethylation of gene body CpG islands predicts high dosage of functional oncogenes in liver cancer. Arechederra M, Daian F, Yim A, Bazai SK, Richelme S, Dono R, Saurin AJ, Habermann BH, Maina F. Nat Commun. 2018 Aug 8;9(1):3164. doi: 10.1038/s41467-018-05550-5.)

The methodology is well constructed, and the experiments performed were adequate to achieve the proposed objective.

The results section is well explained, and the supplementary data allows a better understanding of the results.

The discussion section is clear; however, it has a lack of references to support the results. The authors should perform a more extensive article search and discuss them. This might increase the scientific soundness and interest for the readers.

Overall, in my opinion the article should be accepted for publication but due to the lack of references the authors must perform a more extensive search.
